# Oral Health of Rural Cameroonian Children: A Pilot Study in Bamendou

**DOI:** 10.3390/children10081396

**Published:** 2023-08-16

**Authors:** Guillaume Aimond, Béatrice Thivichon-Prince, Célia Bernard-Granger, Coline Gisle, Tatiana Caron, Andre Valdese Jiokeng, Stefano Majoli, Jean-Christophe Maurin, Maxime Ducret, Laurent Laforest

**Affiliations:** 1Faculté d’Odontologie, Université Claude Bernard Lyon 1, 69008 Lyon, France; beatrice.thivichon-prince@univ-lyon1.fr (B.T.-P.); celia.bernard-granger@etu.univ-lyon1.fr (C.B.-G.); coline.gisle@etu.univ-lyon1.fr (C.G.); tatiana.caron@etu.univ-lyon1.fr (T.C.); jean-christophe.maurin@univ-lyon1.fr (J.-C.M.); maxime.ducret@univ-lyon1.fr (M.D.); laurent.laforest@univ-lyon1.fr (L.L.); 2Pôle D’Odontologie, Hospices Civils de Lyon, 69007 Lyon, France; 3Laboratoire de Biologie Tissulaire et Ingénierie Thérapeutique, Institut de Biologie et Chimie des Protéines, UMR CNRS 5305, 69367 Lyon, France; 4Solidarité Sans Frontières, Yaounde P.O. Box 4260, Cameroon; 5Groupe Médical St-Hilaire (GMSH) Bastos, Yaounde P.O. Box 5123, Cameroon; 6Independent Researcher, 1212 Lancy, Switzerland; 7Département de Prévention et Pathologie Buccale, Division de Stomatologie et Chirurgie Orale, Université de Genève, 1202 Genève, Switzerland; 8Laboratoire des Multimatériaux et Interfaces, UMR CNRS 5615, Université Claude Bernard Lyon, 69622 Villeurbanne, France

**Keywords:** oral health, health promotion, oral hygiene, dental caries, Cameroon, dentistry

## Abstract

Access to dental care in Cameroon is a public health issue, particularly for children living in rural areas. Given the lack of recent data, the investigation of children’s oral health along with their oral hygiene behavior, needs in terms of care, and access to oral hygiene materials were investigated. This cross-sectional pilot study was conducted in Bamendou, Western Region of Cameroon. The study population included 265 children aged 3–18 years who completed a questionnaire about their oral hygiene practices. A clinical examination assessed dental caries, calculus, gingivitis, and oral hygiene. The Chi-squared test was used to identify potential factors influencing caries prevalence rates (significance threshold: *p* < 0.05). Among the 265 children (females: 41.5%, mean age 9.3 years), caries prevalence (ICDAS ≥ 2) was 78.5% and significantly increased with age: 62.2% (3–6 years), 80.9% (7–11 years) and 84.1% (12–18 years, *p* = 0.01). Virtually no children (95.1%) had ever visited a dentist. While only 23.4% of children brushed their teeth at least twice a day, 14% worryingly reported the use of products other than toothpaste (ash, soap, salt, or bicarbonate) and 13.6% no brushing product. The present study revealed a high prevalence of dental caries in this population and inadequate toothbrushing habits, which highlights the need for preventive oral health education and intervention to address these issues.

## 1. Introduction

According to the World Health Organization (WHO), oral diseases are among the most common non-communicable diseases in the African region and can occur throughout life, leading to pain, disfigurement, social isolation, distress, and even death. Dental caries is a common problem with a significant impact on the lives and health of people living in sub-Saharan Africa including Cameroon. The WHO’s response has been to adopt a “Regional Strategy for Oral Health 2016–2025 in Africa”, which aims to halt the progression of dental caries in children and adolescents by 2025 [1]. It is of note that 42% of the Cameroonian population was under 15 years old in 2021 (birth rate = 35 per 1000 women) [2]. Therefore, oral health prevention should target children and teenagers to promote the assimilation of appropriate behaviors at an early stage, and prevent the onset of carious and/or periodontal diseases in a large proportion of the population [3]. The final aim should be to achieve a long-term improvement in the oral health of the population.

To our knowledge, the most recent epidemiological available data on children’s oral health in Central Cameroon dated back to the years 1999 and 2003, and thus needed to be re-evaluated [4,5]. A study carried out in 2003 in the Central Region of the country showed a prevalence of carious lesions of 73.4%, 68.9% and 70.8% among 4–6 years old, 7–11 years old, and 12–18 years old, respectively. Regrettably, Cameroon has only one dental surgeon per 100,000 inhabitants (compared to 66 in France [6]), which is a major barrier for most inhabitants of rural areas to access dental care [7]. Furthermore, there is no state-based oral health program in rural areas, which may result in inadequate oral hygiene education.

Given the absence of recent data on rural Cameroonian children’s oral health status and the absence of appropriate oral care facilities in the vicinity [8], new studies are strongly desirable to document their current oral health condition and the occurrence of any potential changes within the last decades. Furthermore, a good knowledge of children’s hygiene and dietary habits, and needs in terms of dental care, is critical for the implementation of an effective preventive oral health intervention [9].

To address these different objectives, a pilot study was carried out in a rural area of Cameroon to investigate children’s current oral health status and hygiene habits in different age categories, along with the assessment of the local needs in terms of oral care and prevention. Another goal was to verify whether potential changes in caries prevalence were observed compared to the findings reported by previous studies [4,5]. A pilot study was conducted in order to gain detailed knowledge of the local context of rural Cameroon and to identify the potential logistical barriers to overcome in view of conducting a larger study [10]. Our null-hypothesis was the absence of any improvement in children’s oral health and hygiene habits since the preceding studies conducted in rural Cameroon.

## 2. Materials and Methods

### 2.1. Context of the Study

The present pilot study was incorporated into an oral health prevention program, implemented by a French Non-Governmental Organization (NGO): Sourires Sans Frontières [11]. The intervention included children’s oral examination and oral health education sessions. All clinical equipment needed for the clinical intervention was provided by French private dental surgeons and was brought from France. The oral health program included both educational sessions based on illustrated documents and practical exercises. Different topics were addressed: oral cavity and tooth morphology, dental caries development process, nutritional issues in oral health, and oral hygiene methods. Dental jaw models were used to explain toothbrushing technique. These sessions were followed by a giveaway of 2295 toothbrushes and 2354 tubes of toothpaste so that children could put into practice their newly acquired knowledge. Most oral hygiene materials had been brought from France, though some samples of toothpaste had been bought locally to stimulate the sales of oral hygiene products in these rural areas.

### 2.2. Study Design and Studied Population

An official authorization was signed on 4 August 2021 by the sub-prefect of the Penka-Michel district after the creation of a technical organization committee for an intervention from 5 to 15 August 2021 in Bamendou (Cameroon, Western Region, Menoua Department, Penka-Michel District, Dschang). 

Participants of the present cross-sectional pilot study were recruited at the site dedicated to our oral health prevention intervention. A convenience sample was used, consisting of Bamendou children/teenagers aged 3–18 years whose parents accepted their participation in the study. In accordance with the Declaration of Helsinki and the local regulatory guidelines, all subjects or their legal representative gave their informed consent for inclusion before they participated in the study.

### 2.3. Oral Investigations and Data Collection

#### 2.3.1. Dental Interview 

An ad hoc questionnaire inspired from a previous study [5] was specifically designed to assess participant’s oral hygiene, the presence of dental pain, consuming candies and/or sodas, and presence of any past visit to a dental surgeon in the past. Each child/teenager was anonymously identified by number, age, and gender. Given participants’ young age, questionnaires were completed by Cameroonian adult interviewers for enhanced reliability of the answers. Interviewers had previously been trained by the French operators supervising the intervention: four dental students having completed their fourth year of dental surgery from Lyon Dental University, France (GA, CG, CBG, TC).

#### 2.3.2. Dental Examination

An oral clinical examination was then carried out by the French operators for all children with a completed questionnaire. Several data were collected: the number of cavities and their International Caries Detection and Assessment System (ICDAS) grade, the presence of gingivitis (absent/mild/severe), the presence of tartar (absent/moderate/severe), oral hygiene (poor/moderate/good) according to the presence or absence of plaque, and food debris. 

The ICDAS ordinal scale (0–6) measures the severity of dental carious lesions. From Grade 3 onwards, this lesion is irreversible since it has spread beyond the enamel. This marker was chosen for its reproducibility and accuracy in detecting dental carious lesions [12]. The presence of carious lesions was defined using, in parallel, both ICDAS grade ≥ 2 and ICDAS grade ≥ 3 thresholds in the analyses. The total number of lesions in each mouth was also recorded. 

### 2.4. Statistical Analyses

The analyses of this pilot study were essentially descriptive. First, children’s general characteristics, reported toothache, toothbrushing habits, and the different clinical parameters of oral health were described. Then, the prevalence rates of ICDAS grade ≥ 2 dental caries were described, along with the proportion of children presenting ≥ 4 lesions of this grade. Similar analyses were conducted for ICDAS ≥ 3. Then, the caries prevalence rate (ICDAS ≥ 2) and candy/soda consumption were studied according to three age categories (3–6, 7–11, 12–18 years), respectively, representing the successive stages of dental development, namely, stable temporary, mixed, adolescent dentition, and stable young adult dentitions, similarly to a previous study [5]. Qualitative variables were described by percentages and quantitative variables by mean and standard deviation. For inferential analyses (significance threshold: *p* < 0.05), the Chi-squared test was used and in case of non-validity, it was replaced by Fisher’s exact test. Analyses were performed using EPI-INFO 7 ^TM^ version software (Center for Disease Control and Prevention, Atlanta, GA, USA).

## 3. Results

No significant logistical difficulties were encountered in the present study. Among the 288 children whose family accepted their participation in the present study, 16 were excluded because they did not meet the inclusion criteria of age (3–18 years) and seven others because of incomplete or missing data. The analysis therefore included a total of 265 children (females: 41.5%, mean age: 9.3 years SD = 3.1). None of the participants presented any disability or any congenital deformity. 

### 3.1. Oral Health Outcomes and Eating Habits

A total of 31% of children reported the presence of toothache (8.4% “often” or “always”). The overall prevalence of children with at least one decayed tooth (ICDAS ≥ 2) was 78.5% and significantly increased (*p* = 0.01) according to age (Figure 1). A quarter (24.5%) of children had four or more decayed lesions of this grade. Similarly, a total of 68.3% of children presented at least one decayed lesion (ICDAS ≥ 3) and 19.6% at least four decayed lesions (Table 1).

In total, 27.2% of the children presented mild gingivitis and 6.8% severe gingivitis; 41.3% had a moderate amount of tartar on their teeth and 14% had a significant amount; and 28.3% had poor oral hygiene. A total of 98.8% of children reported consuming candy, and 84.7% soda (Table 2). The proportion of soda drinkers significantly increased with age: 68.2% among those aged 3–6 years, 85.3% among those aged 7–11 years, and 95.2% among those aged 12–18 years (*p* = 0.0007), while there was no significant difference for candy consumption (*p* = 0.54). Nearly one in six reported a daily consumption of candy and/or soda (15.6%), with no significant difference between age groups (*p* = 0.41).

### 3.2. Oral Hygiene Habits

Among the 248 children who declared brushing their teeth (93.6%), virtually all of them had a toothbrush (97.6%) whereas others used their fingers. In the total study population, 23.4% brushed their teeth at least twice a day, while 52.5% did so once a day. While most children used exclusively toothpaste (72.3%), others used (sometimes or always) alternative products such as ash, soap, salt, or bicarbonate (14%), or even nothing (13.6%); miswak was never cited. Furthermore, 95.1% of the children had never visited a dentist (Table 2).

## 4. Discussion

The present pilot study presents welcome updated data for children’s oral health in rural Cameroon. These findings highlight in this population of children the persistence of most oral health issues previously identified in rural Cameroon [4,5,13], such as a worrisome prevalence of caries, which increased with age, as well as tooth-related symptoms. In addition to a prevalent consumption of sugary food and drink, the frequency of toothbrushing was inadequate while no local dental care was easily available.

The current study revealed a poor oral health in these children, as evidenced by the high prevalence rates of carious lesions (78.5% with at least one decayed tooth ICDAS ≥ 2) and gingivitis (34%). In parallel, a minority of children brushed their teeth at least twice a day (24.3%), while a worrisome proportion of them used either alternatives to toothpaste (14%), or even nothing (13.6%) during toothbrushing. This is of concern given the absence of fluoride in these alternatives [14]. Furthermore, an inadequate or even absent fluoride concentration in some toothpastes is another issue [15]. These unsatisfactory frequency of toothbrushing could be in part attributable to an insufficient health awareness of families [16].

Some explanations could account for the increasing prevalence of caries with age. First, the appearance of mixed dentition makes toothbrushing technically more difficult for children, resulting in a loss of effectiveness. Then, as children become gradually more autonomous with age, they tend to be more inclined to buy candy and soda on their own, as exemplified by the steady increased rate of soda drinkers according to age in the present study (reaching 95.2% in the oldest age group). Furthermore, in the absence of any local oral facilities, carious lesions may cumulate over time and thus contribute to this increased prevalence with age [17].

Almost the same oral health issues have previously been reported in rural Cameroon [4,5,13] suggesting their persistence over recent decades. Interestingly, different reviews investigating children’s oral health in different age categories in Africa and other developing countries found an overall upward trend of caries prevalence over the last decades, [10,17,18], in part attributable to an easier consumption of sweetened food notably driven by urbanization [10]. It is also of note that the results of the present study are overall consistent with those observed in developing countries other than Cameroon [19], notably in a recent systematic review conducted in the Middle East [20] and in a meta-analysis compiling results from 21 countries worldwide [21]. This may be explained by the numerous barriers that persist in developing countries that prevent any improvement of children’s oral health. Access to dental services and sometimes to appropriate oral hygiene products can be a challenge, often resulting from shortcomings of the national health system [8], particularly in rural and socially deprived urban areas [22], and, when available, financial barriers could jeopardize the affordability of these facilities [23]. The presence of disabilities or congenital deformities could also entail additional difficulties to achieve an appropriate level of oral health [24]. Furthermore, a paramount issue is the prevalent limited parental health awareness, which is highly associated with socioeconomic level and cultural context [25,26]. An insufficient concentration of fluorides in drinking water in some places must not be overlooked either [27]. Nonetheless, even though numerous studies underlined the beneficial impact of fluoridated water on children’s caries incidence [28], this benefit is disputed by other authors [29].

Contrasted differences in children and/or teenagers caries prevalence rates have been observed in similar age categories between and even within countries [10,18,20,21,25]. Besides potential study methodological issues, such differences could also reflect the marked diversity of the investigated populations. For instance, in the review reported by Uribe et al., caries prevalence in pre-school children ranged from 18% to 75% between the 14 studies conducted in different regions of Brazil. The uneven distribution of dental facilities across the country, along with local particularities in cultural, socioeconomic, and/or health-awareness factors could explain these contrasted findings [22,25]. These various factors can also explain potential differences observed in different developing countries regarding children’s dental caries prevalence rate [20]. Thus, an extensive and multidimensional understanding of children’s living context is a mandatory prerequisite before implementing any oral health intervention in developing countries [10].

Interestingly, a recent meta-analysis highlighted the effectiveness of school-based interventions in significantly reducing the burden of oral disease in children living in middle- and low-income countries [30]. Indeed, a sustained intervention is critical to ensure a perennial benefit of educative action. Given their influence on children’s behavior [10,25], involving parents and even teachers in such educative programs is strongly advocated for a more effective implementation of the promoted oral behavior [31]. In parallel, any potential gaps in the access to oral hygiene products or clinical dental care should be filled wherever needed [32]. 

This pilot study presented several limitations, mainly due to its exploratory objectives. For instance, no sample size calculation was computed and a convenience sample was used; furthermore, given the absence of formal calibration between examiners, the distinction between ICDAS grades ≥ 3 was not possible, and ICDAS grade 1 could not be assessed due to the difficulty of drying the teeth on-site. In addition, non-validated scales were used for the other dimensions of oral health investigated; notably, the assessment of eating habits was basic and restricted to candy and soda, leading to an underestimation of exposure to sugar. Nonetheless, there was clearly a worrisome prevalent consumption of sugary food in these children with no access to dental facilities which must be addressed. Another point is that the study was conducted in a small geographical location which can be hardly extrapolated to all rural areas of Cameroon, even though similar trends have been reported in previous studies conducted in rural Africa [4]. In this pilot study, we did not try to assess the local consequences of the COVID pandemic on children’s oral health or any potential changes in oral hygiene habits [33]. Last, only a few factors potentially influencing oral health could be investigated in the pilot study, highlighting the need for future research to explore more exhaustively and with a broader range of variables factors potentially influencing children’s oral health in rural Africa. Despite these limitations, the strength of the present study lies both in the scarcity of such a field study and its paramount importance to conduct appropriate oral health and educative intervention. 

Satisfactorily, no major barriers in terms of logistics, implementation of study protocol, or data collection were encountered during this pilot study. Furthermore, the intervention was favorably welcomed by the local population as participants received a free dental consultation as well as oral hygiene education, for which they were grateful. Based on this successful preliminary experience, a larger study has been implemented in different rural and urban areas of Cameroon, associated with oral health educative sessions. In addition to the use of validated oral indexes, various potential societal and/or behavioral-related factors have been explored; statistical analyses are currently ongoing. 

As a perspective, along with the use of fluoride [34], casein phosphopeptide-amorphous calcium phosphate [35] and biomimetic hydroxiapathite [36] have been reported to be effective in caries prevention. Future investigations are desirable in order to encompass these factors, specifically in these populations of rural African children without any easy access to dental care.

In conclusion, this pilot study provided an initial overview of children’s oral health status in a rural area of Cameroon, highlighting a worrisome prevalence of carious diseases and inadequate oral hygiene behaviors despite acceptable access to toothbrushes. Moreover, in the absence of a state-based oral health program, the findings suggest that the implementation of local specific actions combining educative programs, and provision of dental material and dental care is a key priority, ideally on a regular basis. Indeed, a lack of knowledge and motivation of children and their parents may contribute to the persistence of inappropriate oral health behavior and poor outcomes.

## Figures and Tables

**Figure 1 children-10-01396-f001:**
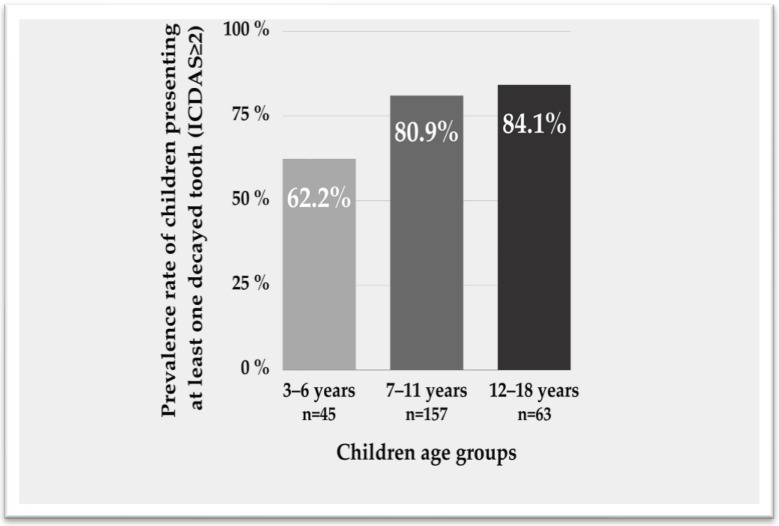
Dental caries prevalence (ICDAS ≥ 2) according to age groups (*p* = 0.01). ICDAS: International Caries Detection and Assessment System.

**Table 1 children-10-01396-t001:** Participants’ oral health characteristics.

	n ^(1)^	%
Number of carious lesions ICDAS ^(2)^ ≥ 2		
At least one	208	78.5
At least 4	65	24.5
Number of carious lesions ICDAS ^(2)^ ≥ 3		
At least one	181	68.3
At least 4	52	19.6
Gingivitis		
None	175	66.0
Mild	72	27.2
Severe	18	6.8
Tartar		
None	118	44.7
Moderate	109	41.3
Severe	37	14.0
Oral hygiene ^(3)^		
Correct	38	14.3
Inadequate	152	57.4
Poor	75	28.3

^(1)^ Counts that do not add to 265 are due to missing values; ^(2)^ The ICDAS ordinal scale (0–6) measures the severity of carious lesions. From Grade 3 onwards, this lesion is irreversible; ^(3)^ Presence of dental plaque and/or food debris.

**Table 2 children-10-01396-t002:** Participants’ declared oral hygiene and eating habits.

	n ^(1)^	%
Frequency of toothbrushing		
None	17	6.4
No daily brushing	47	17.7
Once a day	139	52.5
At least twice a day	62	23.4
Item used for toothbrushing		
Toothbrush	240	97.6
Finger	6	2.4
Other	-	
Products used for toothbrushing		
Toothpaste exclusively	170	72.3
Toothpaste + other products ^(2)^	5	2.1
Other products exclusively	28	11.9
Nothing	32	13.6
Frequency of drinking soda		
Never	40	15.3
Sometimes	194	74.0
Daily	28	10.7
Frequency of eating candy		
Never	3	1.2
Sometimes	243	92.7
Daily	16	6. 1
Daily consumption of candy and/or soda	41	15.6
Ever visited a dentist		
Yes	13	4.9
No	250	95.1

^(1)^ Counts that do not add to 265 are due to missing values; ^(2)^ Ash, soap, salt, or bicarbonate.

## Data Availability

The datasets used and/or analyzed during the current study are available from the corresponding author upon reasonable request.

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
