# Peer review of "Oral Health of Rural Cameroonian Children: A Pilot Study in Bamendou"

_children, 2023, doi:10.3390/children10081396_

Round 1
Reviewer 1 Report
Specific Comments
Title
Suggested title: Oral Health of Cameroonian Children: A Pilot Study in Bamendou
Abstract
Needs English language editing and revision.
The abstract should be a single paragraph and should follow the style of structured abstracts, but without headings.
Please add method of statistical analysis and level of significance.
Also, please add calculated P-values.
P-value NOT p-value.
Keywords: Please use keywords from the MeSH Database.
Introduction
Please add more than 1 reference for each paragraph.
At the end of the Intro section, please give your null hypothesis. The latter should be derived from the preceding thoughts in this section and should be broached again in the Discussion. In hypothesis testing, the null hypothesis is the one you are hoping that is can be disproven by the observed data.
Materials and Methods
Needs more elaboration.
Ethical approval is missing. Also, please add approval number.
Please mention sample size calculation.
What is the study rationale?
Please write the study design.
Was there a control group?
Please add abbreviation following full term when possible. The first time an abbreviation appears, it should be placed in parentheses following the full spelling of the term.
Please add the name of the manufacturer for all materials and equipment used.
P-value NOT p-value.
Results
P-value NOT p-value.
Was any kind of calibration done? Reliability needs to be clarified (intra or inter or both).
Please mention kappa coefficient value.
In figures 1 and 2, please clarify abbreviations. A footnote explaining the abbreviations need to be added. Also, please in both figures, you need to be clearer regarding the labels of X and Y axis.
Discussion
This section may usefully start with a summary of the major findings, but repetition of parts of the abstract or of the results section should be avoided.
“In the present/current study” NOT “In this/our study”.
Please mention future directions.
Please point out the implications of the findings and their limitations.
Conclusions
Appropriate.
References
Also, please check journal guidelines for reference writing.
References needs to be 10 years back not more (from 2012 to 2022).
Old references need to be replaced by recent ones.
Some of the references include DOI, others do not include DOI number.
In general, all references need to be revised, standardized and written according to the journal guidelines.

Moderate editing of English language required
Author Response
We thank the reviewer for these helpful comments. We did our best to address them. Our replies in the present the document and the corresponding changes in the manuscript are highlighted in red.

Reviewer 2 Report
Oral Health of Cameroonian Children: A Pilot Study in Rural Areas
What is the originality of the study?
The questionnaire should be presented.
Figure 1. Dental caries prevalence can be omitted
The discussion does not focus on the results, which show dental caries prevalence and oral hygiene habits.
This paper does not bring any new and original data to the literature.
Some minor spelling issues should be resolved.
Author Response

(The authors gave the same response as above.)

Reviewer 3 Report
Please mention if ethical approval was obtained.
They mention that French operators did this research on Cameroon children. How long they did the study is not mentioned. So they need to mention the time frame.
Please add more on the sample selection and sample size selection.
More discussion is needed to compare the dental caries in children with children in other countries in developing nations.
https://www.researchgate.net/publication/339137069_Prevalence_of_Dental_Caries_and_Oral_Hygiene_Practice_in_School_Children_of_Bhaktapur_Nepal
https://head-face-med.biomedcentral.com/articles/10.1186/s13005-020-00237-z
https://www.emro.who.int/emhj-volume-26-2020/volume-26-issue-6/prevalence-of-dental-caries-among-children-aged-515-years-from-9-countries-in-the-eastern-mediterranean-region-a-meta-analysis.html
A minor check of English is required.
Author Response

(The authors gave the same response as above.)

Reviewer 4 Report
Dear Authors, thank you for the interesting article to review. Here are some suggestions of mine:
1. In lines 41-42, please add the information on birth rate
2. There is no information, wether the methodology was prepared according to the OHIP standards - if so, wether the standards were prepared to this speciffic study, validated for the Cameroonian children. Please, add that information in M&M.
3. To my mind, the group age should be "children and young adults", as 18 years is an adult already
4. Line 153+ - please modify. Caries is not an indicator of poor oral hygiene, rather a result.
5. The results should be compared not only to the previous studies on Cameroonian population, but also it would be valid to add to the discussion other African countries, as well as western countries
6. Did the Covid pandemic influence the oral health habits of Cameroonian population - please, discuss that, see the similar results of other countries:
Varkey IM, Ghule KD, Mathew R, et al. Assessment of attitudes and practices regarding oral healthcare during the COVID-19 pandemic among the parents of children aged 4–7 years. Dent Med Probl. 2022;59(3):365–372. doi:10.17219/dmp/147184
7. In the results and discussion, please add if any patient was somehow disabled or had any congenital deformity - those are the factors that influence the special care needs and might require more dental care needs, as and example, please see cleft patients
- Paradowska-Stolarz A, Mikulewicz M, Duś-Ilnicka I. Current Concepts and Challenges in the Treatment of Cleft Lip and Palate Patients-A Comprehensive Review. J Pers Med. 2022 Dec 19;12(12):2089. doi: 10.3390/jpm12122089.
8. Please, add the strong and weak points of the study (limitations).
The language needs improvements in punctation, please check that
Author Response

(The authors gave the same response as above.)

Reviewer 5 Report
Dear Authors,
I have read the manuscript with interest and some questions raised. Enlisted please find my comments.
Overall. General English grammar revision (Minor spelling errors).
Key words. “dentistry” could be added in my opinion.
Abstract. Please add the names of the statistical tests in this section.
Materials and Methods. For each material used, please add details about commercial name manufacturer, City and State.
Materials and Methods. For each machinery used, please add details about commercial name manufacturer, City and State.
Materials and Methods. Please add details about software used, version, Manufacturer, City and State.
Results. Authors stated “The analysis therefore included a total of 265 children (females: 41.5%, mean age: 9.3 years SD=3.1” Please add in materials and methods section if and how sample size calculation has been performed
Discussion. Authors stated “. Last, only few factors potentially influencing oral health could be investigated in the pilot study.”. Provide a general interpretation of the results in the context of other evidence, and implications for future research. It could be added that “Future research is needed in order to take into careful account other variables. In fact, the use of fluoride (Historical and bibliometric notes on the use of fluoride in caries prevention. Zampetti P, Scribante A. Eur J Paediatr Dent. 2020 Jun;21(2):148-152.), casein phosphopeptide-amorphous calcium phosphate (Efficacy of a Novel Bioactive Glass-Polymer Composite for Enamel Remineralization following Erosive Challenge. Fallahzadeh F, Heidari S, Najafi F, Hajihasani M, Noshiri N, Nazari NF. Int J Dent. 2022 Apr 22;2022:6539671. doi: 10.1155/2022/6539671..) and biomimetic hydroxiapathite (Biomimetic hydroxyapatite paste for molar-incisor hypomineralization: A randomized clinical trial. Butera A, Pascadopoli M, Pellegrini M, Trapani B, Gallo S, Radu M, et al. Oral Dis. 2022 Sep 22. doi: 10.1111/odi.14388.) have been introduced and showed significant results in caries prevention. Future reports are needed in order to encompass also these variables”. These concerns should be added to Discussion section.
Discussion. Please add a comparison between the results of the present report and the results obtained in other studies.
Tables. None presented. Please add some tables showing the results and descriptive statistics of the main variables tested.
References. Some references are quite old (1999;1994;1996;). If possible, please switch with some more modern research. Some recent studies have been suggested in the sections above.
English needs just some minor revisions
Author Response

(The authors gave the same response as above.)

Round 2
Reviewer 1 Report
None
None
Author Response
Many thanks for your helpful comments. This enabled us to significantly improve our manuscript. Best regards.
Reviewer 2 Report
The manuscript does not bring any new elements to the body of knowledge.
"What is the originality of the study? This paper does not bring any new and original data to the literature.
Reply to reviewer: We acknowledge that the present study does not bring any novelty compared to the preceding studies. A noticeable issue is their scarcity: the most recent one conducted in rural Cameroonian areas dates back from 2001-2002. Consequently, updated data were critically needed. Furthermore, these updated results are of major interest since they worryingly highlight the absence of any change during this long period. Lastly, as discussed in the manuscript, such surveys are critically needed in view of implementing appropriate oral health intervention programs "
Author Response
Thank you for your reply. Even though some similar studies have been previously conducted in sub saharian Africa, we keep believing that our findings remain of interest given the critical need of data on African children’s oral health. Further, our manuscript has been enriched with more details on the intervention conducted in these villages.
Have a nice Summer.
Best regards
Reviewer 5 Report
All comments have been answered. Thank you
Author Response

(The authors gave the same response as above.)
